# Analysis of the Genetic Parameters for Dairy Linear Appraisal and Zoometric Traits: A Tool to Enhance the Applicability of Murciano-Granadina Goats Major Areas Evaluation System

**DOI:** 10.3390/ani13061114

**Published:** 2023-03-21

**Authors:** Javier Fernández Álvarez, Francisco Javier Navas González, Jose Manuel León Jurado, Carlos Iglesias Pastrana, Juan Vicente Delgado Bermejo

**Affiliations:** 1National Association of Breeders of Murciano-Granadina Goat Breed, Fuente Vaqueros, 18340 Granada, Spain; 2Department of Genetics, Faculty of Veterinary Sciences, University of Córdoba, 14071 Córdoba, Spain; 3Centro Agropecuario Provincial de Córdoba, Diputación Provincial de Córdoba, 14071 Córdoba, Spain

**Keywords:** does, local breed, conservation, adaptability, variance components, heritability, genetic and phenotypic correlations

## Abstract

**Simple Summary:**

Murciano-Granadina has drifted towards better dairy-linked conformation traits without losing the grounds of the zoometric basis which confers it with an enhanced adaptability to the environment. Consequently, international Linear Appraisal Systems (LAS) may not fit the zoometric variability reality of autochthonous breeds, such as the Murciano-Granadina goat. LAS panels comprise large numbers of traits which makes selection for dairy conformation a complex and time costly task, hence selection practices focusing on major areas is often suggested. The evaluation of genetic, phenotypic and environmental parameters for each zoometric/LAS trait individually and of the pairwise relationships among traits may permit the design of a solid selection strategy towards the maximization of dairy potential while making selection tasks time- and resource-efficient. Results suggest that zoometrics and LAS-derived genetic and phenotypic parameters are translatable as long as the process of collection is performed objectively by trained operators. Selection of major areas is feasible but may be conditioned to the restructuration and modification of the scales that are currently used for dairy goats. The strategies that were designed help to evaluate the momentum of selection for dairy-linked zoometric traits of the Murciano-Granadina population and its future evolution to enhance the profitability and efficiency of breeding plans.

**Abstract:**

Selection for zoometrics defines individuals’ productive longevity, endurance, enhanced productive abilities and consequently, their long-term profitability. When zoometric analysis is aimed at large highly selected populations or in those at different levels of selection, linear appraisal systems (LAS) provide a timely response. This study estimates genetic and phenotypic parameters for zoometric/LAS traits in Murciano-Granadina goats, estimating genetic and phenotypic correlations among all traits, and determining whether major area selection would be appropriate or if adaptability strategies may need to be followed. Heritability estimates for the zoometric/LAS traits were low to high, ranging from 0.09 to 0.43, and the accuracy of estimation has improved after decades, rendering standard errors negligible. Scale inversion of specific traits may need to be performed before major areas selection strategies are implemented. Genetic and phenotypic correlations suggests that negative selection against thicker bones and higher rear insertion heights indirectly results in the optimization of selection practices in the rest of the traits, especially those in the structure, capacity and mammary system major areas. The integration and implementation of the strategies proposed within the Murciano-Granadina breeding program maximizes selection opportunities and the sustainable international competitiveness of the Murciano-Granadina goat in the dairy goat breed panorama.

## 1. Introduction

Linear Appraisal Systems (LAS) were developed as time cost-effective alternatives to provide a timely relatively accurate response to the need to perform large scale zoometric evaluations. The biggest problem associated with in vivo zoometry (measurement and comparison of the sizes and proportions of animals or animal parts) is the difficulty of restraining animals in a natural position long enough to make an accurate measurement, especially when the differences being measured are small [1].

In this context, the complementarity between LAS and zoometry must be performed at an acceptable repeatability level across appraisers. This means that it must be possible to define the trait and all its components and the associated evaluation criteria precisely enough for the trait to be evaluated by appraisers with acceptable repeatability.

Despite few preliminary applications [2], the National Association of Breeders of the Murciano-Granadina goat breed (CAPRIGRAN) [3] started routinely implementing its LAS in 2010 [4]. Since then, CAPRIGRAN LAS has assisted Murciano-Granadina goat breeders in the evaluation of their individual animals based on (morpho) type traits that affect dairy structural and functional durability such as udder measurements [5]. This translated into a ten-times higher increase in the number of Murciano-Granadina LAS evaluations than those implemented in other breeds over the past decade. The implementation efficiency of CAPRIGRAN LAS has been maximized through the integration of the association within “Cabrandalucía”, the Andalusian Federation of Purebred Goat Associations, on 24 February 2005 as an initiative to share the particular projects of each goat breeders association of in Andalusia (Spain).

In these regards, Cabrandalucía Federation has implemented the concept of smart farming, relying on a precision livestock farming (PLF) platform (Web-App RUMIA). Web-App RUMIA incorporates PLF-like principles based on the integration of individual animal data points to optimize decision-making through a smartphone-based terminal and substitutes the previous “Escardillo” software used by CAPRIGRAN [4]. The improvements in the collection of zoometric/LAS information rely on the systematic remote on-farm individual data recording and acquisition, storage processing and interpretation by a Cabrandalucía supercomputer [6].

The productive levels that the Murciano-Granadina breed has reached currently provides it with a prominent position within the international dairy goat breed panorama [7]. In this sense, Murciano-Granadina consideration among the most highly productive dairy goat breeds explains the strong roots of CAPRIGRAN LAS being found in The American Dairy Goat Association (ADGA) and U.S. Department of Agriculture (USDA)’s LAS [7].

CAPRIGRAN LAS routinely comprises the numerical description of 17 zoometric linear traits (10 in the case of bucks as the mammary system major category is not evaluated) on a one-to-nine-point scale to represent the zoometric linear biological range for each particular trait [4]. The term “linear” in a linear appraisal system refers to the fact that traits are rated on a linear scale that goes from one biological extreme for that trait to the other. For primipara and multipara does, the 17 zoometric traits are sorted into four major categories (structure and capacity, dairy structure, mammary system and legs aplomb). Parallelly, for bucks, young males and goats that have not yet given birth, thus have not freshened, the mammary system is not evaluated, hence, only three major categories are considered into which 10 zoometric linear traits are sorted. The same scale is used in males and females for the body depth traits from the structure and capacity major category, and the major categories of dairy structure and legs and feet. Afterwards, linear trait data are comprehensively used to build individual reports for every doe and buck.

One of the main drawbacks derived from the extrapolation of ADGA LAS to build CAPRIGRAN LAS, relies on the scarce amount of information which exists on the heritability of structural traits in dairy goats. Indeed, although genetic parameters may be similar, according to relative indications and experience, the absolute heritability of traits is not known or expected to be the same for dairy cattle and dairy goats. For example, the heritabilities used in the selection of traits for ADGA LAS are based on 4 years of dairy cattle linear data. Hence, they were inferred in other species, and specifically barely any information is present in regards the genetic correlations across zoometric or LAS traits.

Selection for zoometrics defines an individual’s productive longevity, endurance, enhanced productive abilities and consequently, its long-term profitability [8,9]. When zoometric analysis is aimed at large highly selected populations or in those at different selection momentums, LAS may provide a timely selection response. However, the particular selective context of the breed must be evaluated. The particular analysis of each variable permits tailoring specific strategies for each trait and serves as a model for other breeds, either selected or in terms of selection. For CAPRIGRAN LAS to be deemed effective, zoometric and LAS-computed genetic parameters must be comparable, but also, they must be heritable (genetically-controlled) enough (heritability of 0.15 or higher is accepted as indicating at least moderate heritability of a trait [10]) so that progress or improvement can be made at an acceptable rate through the selection of sires. Traits that are not at least moderately heritable are more effectively handled through herd management practices (such as culling) and are not suitable for inclusion in LAS.

Recent studies have suggested Murciano-Granadina goats’ inherent highly rustic nature may make CAPRIGRAN LAS not to optimally meet the ADGA and USDA’s LAS standards (upon which CAPRIGRAN LAS was formerly based) [5]. Contextually, this may hinder the efficiency of selection practices focused towards the enhancement of dairy-linked conformation in Murciano-Granadina goats, while sustainably maintaining its increased productivity under the harsh conditions where the breed was traditionally bred.

Contextually, CAPRIGRAN international disagreement compelled the evaluation, optimization, validation and suggestion of restructuration measurements for CAPRIGRAN LAS [4,5]. Recently, studies validated CAPRIGRAN LAS solidity and internal consistency for the measurement and capture of the variability of dairy-related zoometric parameters [5]. However, optimization was limited, given that resulting models were quite conservative, with only rump angle lacking representativeness in the explanation of milk yield and milk quality-related traits [4]. Among other proposals, researchers suggested a limb-related traits scale levels reduction/readjustment, bucks’ stature extension and male category age group subdivision (bucklings younger than two years and bucks of two years old and older) as the most relevant modifications to implement to adapt CAPRIGRAN LAS scales being applied to the reality described by the individuals of the Murciano-Granadina breed [5].

The value of CAPRIGRAN LAS relies in the possibility of dairy goat breeders using the information provided by animal evaluation programs as guidance in making their management decisions, such as mating plans that involve the selection of sires or dams used in their breeding programme [11]. In turn, these management decisions may not only influence the structural correctness and genetic potential of individual animals, which determines their lifetime in the herd and their overall production level, but also may help with understanding how the condition of type traits affects the structural durability and the reproductive and production efficiency of an animal is critical to effective herd management. This means dairy goat herds evaluated with the LAS will be instrumental in helping develop the data base needed to determine structural traits heritability in dairy goats and, eventually, their relationship to longevity and production, thus, their economic value.

To this aim, this paper determines whether phenotypic, genotypic and environmental parameters for traditional zoometric analysis and CAPRIGRAN LAS are comparable. Afterwards, the appropriateness of the clustering of zoometric or LAS items comprising each mayor category will be tested to propose enhancement measures to ensure the potential of Murciano-Granadina breed selection strategies is maximized. This may help to evaluate the viability of selection strategies based on the relationship across zoometric and LAS traits as a base for future studies evaluating potential benefits linked to an increased productive longevity.

## 2. Materials and Methods

### 2.1. Animal Sample and Linear Appraisal Records

The Murciano-Granadina whole pedigree datafile comprised 279,264 animals (266,793 does and 12,971 bucks), and was used as the pedigree matrix for genetic analyses. Animals had been born from June 1966 to November 2019. The linear appraisal had been historically performed in 41,418 animals. Animal records were collected from 76 farms (Median: 417; Min: 2; Max: 3402) in the South of Spain from 9 June 2010 to 18 December 2019. All the farms considered in the study had received official National and International Sanitary Certificates. All farms were controlled and officially declared tuberculosis-free (C3), brucellosis-free (M4) (Order of 22 June 2018 [12] and Directive 91/68/EEC) and SCRAPIE RC (Regulation (EC) No 999/2001 of the European Parliament and the Council). These farms also followed voluntary control plans for Caprine Contagious Agalactia (CCA) (National CCA Surveillance, Control, and Eradication Programme 2018–2020) and Caprine arthritis encephalitis (CAEV) (Order AYG/287/2019 of 28 of February of 2019 [13]). Goats were clinically examined by an official veterinarian and individuals presenting signs of illness or disease conditions were officially declared and removed from the herds, hence, discarded from the analyses. Permanent stabling practices were followed by all farms considered, *ad libitum* water, forage and supplemental concentrate were provided. A detailed description of the analytical composition of the diet supplied to the animals is reported in Appendix A.

Records from 95 individuals were discarded due to their zoometric and linear appraisal observations being missing or incomplete. A total of 41,323 records, belonging to 22,727 herdbook-registered primipara does, 17,111 multipara does and 1485 bucks were considered in the analysis. Average ages for primipara, multipara does and bucks in the sample were 1.61 ± 0.35 years, 3.96 ± 1.74 years and 2.43 ± 1.49 years (μ ± SD), respectively.

### 2.2. Murciano-Granadina Linear Appraisal System (LAS)

Each observation comprises each animal’s rater score in the following four major categories for primipara and multipara does (three for bucks, young males and yet-to-give-birth goats); structure and capacity, dairy structure, mammary system (except in males) and legs and aplomb. In primipara and multipara does, each record comprised information on 17 linear traits rated on a 9-point scale. Bucks were not scored for the mammary system major category, hence only 10 9-point scale traits were scored with the aforementioned animals. Body depth from the structure and capacity major category and the dairy structure and legs and feet major categories followed the same criteria independent of sex and sexual status. The same trained rater scored all animals. Once all major categories were scored, the final score represents how close the overall animal comes to the optimal dairy standard. Murciano-Granadina LAS establishes that each major category contributes to the final score based on 25% for structure and capacity, 15% for dairy structure, 20% for legs and feet and 40% for mammary system for primipara/multipara does (any doe producing milk). In bucks and young males, such percentages change to 50% for structure and capacity, 20% for dairy structure and 30% for legs and feet.

Rater’s scores are assigned one of the six category qualifications considered as follows: insufficient (IN) for animals displaying less than 69% of the optimal standard for Murciano-Granadina dairy goats (a final score of 69 points or less), mediocre (R), 70 to 74% of optimal standard (70 and 74 points), good (B) from 75 to 79% of optimal standard (75 to 79 points), quite good (BB) from 80 to 84% of optimal standard (80 to 84 points), very good (MB) from 85 to 89% of optimal standard (85 to 89 points), or excellent (E) when at least 90% of optimal standard is displayed (final score higher than 90 points). The scales used and the translation process from zoometric traits to LAS traits is described in detail in Sánchez Rodríguez et al. [11], Table 1 and Appendix A.

Age elements, such as doe age or lactation order, affect the dairy linear or type appraisal-related traits [14]. These elements, often registered for does at appraisal, permit adjusting models for the outputs of linear or type appraisal records [15,16]. The Pearson product-moment correlation coefficient between lactation stage and age in years was 0.705 (*p* < 0.01), suggesting redundancies if both age components were simultaneously considered. Thus, lactation stage was considered and results for primipara and multipara goats were reported separately.

Common parametric assumptions were tested in Murciano-Granadina goat breed zoometric and LAS historical records collected until December 2019. Kolmogórov–Smirnov and Levene tests were used to evaluate normality and homoscedasticity, respectively, using SPSS Statistics for Windows statistical software, Version 25.0. Given the large sample size used in this study, the nonparametric method to test for the independence of two random variables with continuous distribution function (df) proposed by Hoeffding [17] which uses joint ranks was chosen. To this aim, the *Hmisc* package’s *hoeffd* function [18] of RStudio 1.1.463 by the R Studio Team [19] was used. *p*-values are approximated by linear interpolation on the table in Hollander and Wolfe [20], which uses the asymptotically equivalent Blum–Kiefer–Rosenblatt statistic. For *p* < 0.0001 or >0.5, *p* values are computed using a well-fitting linear regression function in log *p* against the test statistic.

### 2.3. Multicollinearity Testing of Fixed Effects (Factors) and Covariates

To determine the environmental background affecting zoometric and LAS traits, we chose the following set of independent factors (kidding month and season, farm, sex, lactation stage) and covariates of age at kidding (in years) and days in milk following the premises that are commonly found in the literature for the same purpose [15,21,22]. Additionally, we considered the effect of the interaction between farm/kidding year and farm/kidding year/kidding season to verify whether using a linear model to evaluate environmental effects would be appropriate. Redundancies in the variables used were identified after performing the multicollinearity assumption prior to further analyses. VIF was computed using the *Multicollinearity statistics* routine of the *Describing data* package of XLSTAT 2014 (Pearson Edition).

### 2.4. Analysis of Covariance (ANCOVA) Test for Fixed and Random Effects

After multicollinearity testing, we used the ANCOVA from the family of Generalized Linear Models to determine how zoometric and LAS traits vary across independent factors, covariates and interactions. ANCOVA was performed using the *ANCOVA* routine of the *Modelling data* package of XLSTAT 2014 (Pearson Edition). The independent factors considered were as follows: parturition month and season, farm, sex, lactation stage (all qualitative variables that take value form). The covariates of age (in years) and days in milk and the interaction between farm/kidding year were also considered. ANCOVA was run to verify the appropriateness of a linear model comprising the aforementioned environmental effects to explain the variability in zoometric and LAS traits in Murciano-Granadina does and bucks.

A Spearman’s rho (ρ) correlation test must be performed to rule out monotonic redundancies. The Spearman correlation between two variables is equal to the Pearson correlation between the rank values of those two variables except for the fact that while Pearson’s correlation assesses linear relationships, Spearman’s correlation assesses monotonic relationships (whether linear or not). Hence, Spearman’s rho (ρ) explains how well the relationship between two variables can be described using a monotonic function [23].

When an independent variable is related to another independent variable at a correlation of ≥|0.5|, statistical redundancies are detected, hence a model comprising both will not adjust the dependent variable over the relationship between both independent variables. Hence, one or the other should be removed. The decision on which to discard must be made considering the relationship of each independent variable in the pair with the rest of independent variables considered in the model [24].

#### 2.4.1. Fisher’s F Test

The Fisher’s F test was used to examine whether explanatory variables satisfactorily explain the behaviour of zoometric or LAS traits. *p*-values lower than 0.0001 imply there is a lower than a 0.01% risk in assuming that the null hypothesis is wrong (this is not an effect of the explanatory variables).

#### 2.4.2. Goodness of Fit

The R^2^ (coefficient of determination) indicates the percentage of the variability of the dependent variable which is explained by the explanatory variables that remained after multicollinearity analyses. The closer to 1 the R^2^ is, the better the fit of the set of explicative variables is to describe the variability in the respective dependent zoometric trait. The Predicted R-squared decreases when insignificant or redundant terms are added to a particular model (Adjusted R^2^). As a rule of thumb, adjusted and predicted R-squared values should not differ by more than 0.2. According to StatEase [25], there is not a commonly used reference value for R-squared. When a model is significant (*p* < 0.05), there is no evidence for an insignificant lack of fit (*p* > 0.05), and good agreement between adjusted and predicted R^2^. When agreement between Predicted and Adjusted-R^2^ exists, precision is adequate (>4) and the residuals could be presumed to statistically behave well. Hence, the model being tested provides good predictions for outcomes on average. In these regards, low R-squared values are indicative of a certain fraction of individual variation not being explained by the model tested.

#### 2.4.3. Type III Sum of Squares Analysis and Model Predictive Potential

To determine the amount of information provided by each fixed effect and covariate, we evaluated the Type III Sum of Squares SS tables (Table 2). The evaluation of model predictive potential, the significance level and confidence intervals for each level of each parameter were evaluated. Confidence intervals including zero and significance levels *p* > 0.05 are indicative of a statistically non-significant weak impact of that factor on the specific zoometric or LAS traits tested.

#### 2.4.4. Analysis of Residuals

Given the assumptions of the linear regression model evaluated in ANCOVA, standardized residuals should normally distribute, which implies 95% of the residuals should be in the interval [−1.96, 1.96], with all the observations falling outside this interval potentially being outliers, or indicative of the normality assumption not being met. The *DataFlagger* routine of the *Tools* package in of XLSTAT 2014 (Pearson Edition) was used to graphically represent residuals. When the percentage of the residuals that are not in the [−1.96, 1.96] interval exceeds 5% (*p* > 0.05), the analysis could lead to rejecting the hypothesis of normality of residuals, which would render ANCOVA outputs invalid for conclusions to be issued.

### 2.5. Genetic Analyses

#### 2.5.1. Model and Genetic Parameter Estimation for Zoometric and LAS Traits

The complete kinship matrix used for genetic analyses comprised all the 279,264 animals (266,793 does and 12,971 bucks) in the Murciano-Granadina goat breed pedigree. As the literature suggests, when bucks start rutting, that is, when male goats display the behaviours associated with the urge to breed, they go through physical changes which even alter specific variables, such as a rump angle decrease of 3 degrees [26]. Most of goat breeds’ breeding season extends from August to January and males go into rut during Autumn. The rut is characterized in bucks and the males of other species by an increase in testosterone, exacerbated sexual dimorphisms and increased aggression and interest in does [27]. These cyclic changes throughout the year are the source for natural discrepancies in the definition and specific characteristics of zoometric traits between bucks and does, whose body changes are rather progressive along their lives across lactation stages. This in turn may lead to statistical biases, hence, we decided the phenotype dataset should only comprise those observations belonging to does, either primipara or multipara, to estimate genetic and phenotypic parameters. As a result, a total of 39,838 records, belonging to 22,727 herdbook-registered primipara does and 17,111 multipara does were considered in the genetic analysis. Animals were only scored once in their lifetime. Therefore, a multitrait animal mixed model with single measures was used to estimate (co)variance components, and the corresponding heritability (the proportion of the variation in a given trait within a population that is not explained by the environment or random chance, and measures the fraction of phenotype variability that can be attributed to genetic variation), repeatability, phenotypic and genetic correlations (the proportion of variance that two traits share due to genetic causes, the correlation between the genetic influences on a trait and the genetic influences on a different trait estimating the degree of pleiotropy or causal overlap) and standard errors of such correlations for the traits under examination. In matrix notation, the following multitrait animal model with single measures was used: Y_ijklmn_ = μ + Far_i_ · A_i_ + LacStat_j_ · B_j_ + KMon_k_ · C_k_ + IntFarm/KYear_l_ · D_l_ + b_1_DIM_m_ · E_m_ + b_2_A_n_ · F_n_ + $b_{3}^{2}A$
_n_ · F_n_ + e_ijklmn_,where Y_ijklmn_ is the vector of observations for each separate measure of each zoometric or LAS trait (Table 1) for a given animal;μ is the overall mean;Far_i_ is the vector for the fixed effect of the ith farm/herd (i = 76 farms);LacStatj is the vector for the fixed effect of the jth lactation stage (j = primipara/multipara does);KMon_k_ is the vector for the fixed effect of the kth kidding month (k = January to December);IntFarm/KYear_l_ is the vector for the fixed effect of the lth level of interaction between farm/herd and kidding year (l = 400 interaction levels possibilities combining the 76 farms and kidding years from 2005 to 2019);days in milk was considered a linear covariate, hence b_1_ is the linear regression coefficient on days in milk (DIM_m_);age in years was considered a linear and quadratic covariate, hence b_2_ and $b_{3}^{2}$ are the linear and quadratic regression coefficients on the age of evaluation (A_n_);e_ijklmn_ is vector of random residual effects;and A_i_, B_j_, C_k_ and D_l_ are incidence matrices relating records to their respective fixed effects, while E_m_ and F_n_ are incidence matrices relating records to their respective random effects.

Only the direct genetic effect (animal) was fitted in each model due to because zoometrics/LAS scores were recorded only once on each individual animal.

The MTDFREML software package [28] was used to perform restricted maximum likelihood approach-based univariate analyses in order to compute heritabilities and variance components. The same software was used to carry out bivariate analyses to estimate covariates and genetic and phenotypic correlation. Genetic and phenotypic correlations between each individual conformation trait were estimated using a multivariate analysis including all traits. The iteration process used sought a convergence criterion level of 10^−12^. Link functions can be found in Boldman et al. [28]. The standard errors for heritability and genetic and phenotypic correlations were computed using MTDFREML software package [29] as well.

As suggested by Navas González et al. [29], we used the phenotypical variance of each character and the existing phenotypical correlations between each possible pair combination for the estimation of the starting point to seek for the convergence of additive genetic variance component (multiplying them by 0.2). Then, we did the same for environmental variances (multiplying them by 0.8) and genetic and phenotypic correlations to obtain specific variance components and estimates of fixed and random effects for each trait in multivariate analyses. To build the matrix of covariates among zoometric and LAS traits, respectively, the *Bivariate* routine of the *Correlate* procedure of the *Analyze* package in SPSS Statistics for Windows statistical software, Version 25.0. was used. Starting values for genetic, phenotypic and environmental variates and covariates are shown in Appendix A.

#### 2.5.2. Non-Genetic Factors Estimation (BLUES)

After convergence was reached, we directly estimated non-genetic factors estimators through the best linear unbiased estimators for fixed effects (BLUES) using the MTDFREML software [29].

## 3. Results

### 3.1. Preliminary Assumption Testing in Zoometric and LAS Traits

After the study of the distribution and symmetry properties of zoometric traits and the scale readjustment proposal suggested in Fernández Álvarez et al. [11]. Parametric assumptions were met (normality, heteroscedasticity and sample independence, *p* > 0.001) which was supported by the values for skewness statistics ranging from—½ to ½, which evidenced the symmetry of the profile of the curve described by the distribution of the data for all the variables evaluated. According to the evaluation of kurtosis, most of the variables presented a distribution with kurtosis < 3 (excess kurtosis < 0) or platykurtic with low and broad central peaks and short thin tails. Exceptionally, a distribution with kurtosis >3 (excess kurtosis > 0) or leptokurtic was reported for motility of movements in bucks.

### 3.2. Multicollinearity Testing of Fixed Effects (Factors) and Covariates

Variance inflation factor evaluation suggested the model was free from redundancies after two rounds of multicollinearity analyses were performed (Appendix A). Multicollinearity evaluation suggested the need to discard kidding year and the interaction between farm–kidding year–kidding season from further analyses (VIF > 5). Additionally, Spearman’s rho correlation (ρ ≥ |0.5|) denoted a strong monotonic relationship between sex and kidding season with the rest of variables, hence, XLSTAT 2014 (Pearson Edition) automatically discarded them the set of environmental factors and covariates used for the following ANCOVA procedure.

### 3.3. Analysis of Covariance (ANCOVA) Test for Fixed and Random Effects

As suggested in Table 2, we can therefore conclude that all the factors and covariates explain a significant amount of the information contained in zoometric and LAS traits.

Table 3 presents the goodness of fit coefficients of the model. In this particular case, from 10.2 to 70% (9.2 to 69.6% when adjusted) of the variability across zoometric/LAS traits is explained by the independent factors of kidding month and season, farm, sex, lactation stage, the covariates of age at kidding (in years), days in milk and the interaction between kidding year and farm. The remainder of the variability may be ascribed to additional effects (other explanatory variables) not considered during this experiment, for instance, genetic or those related to the nutritional status of the animals as suggested in the literature [30,31].

Significance levels over 0.05 were reached and 0 was not contained in the confidence interval for almost all the levels within all fixed effects and covariates considered in the model (Appendix A). This denoted that all the elements brought relevant information to the explanation of the behaviour of zoometric and LAS traits.

The comparison between predicted values and observed values suggested 5% of standardized residuals ((35,196 × 100)/692,096) could be identified as potential outliers. Hence residual normality assumption was met and ANCOVA outputs can be used to draw valid conclusions.

### 3.4. Genetic Analyses

#### Genetic Model Comparison, Phenotypic and Genetic Parameters Estimation

Estimates of non-genetic fixed effects (BLUES) obtained from the REML quantitative genetic analysis, including days in milk as a linear covariate and age as a linear and quadratic covariate, the fixed effects of farm/herd, lactation stage, kidding month, and the interaction between farm/herd and kidding year are shown in Appendix A. The estimates for heritability, genetic, phenotypic and environmental variance obtained through REML methods for zoometric and LAS traits are shown in Table 4. The genetic (r_G_) and phenotypic correlations (r_P_) estimated are shown in Table 5. Results for zoometric and LAS traits were exactly the same.

Genetic correlations ranged from −0.010 to 0.870, with the highest positive genetic correlation occurring between Mobility and Rear Legs Rear View (0.870) and the lowest positive genetic correlation between Udder Depth and Rump Width (0.000). The most negative genetic correlation (−0.570) occurred between udder depth and anterior insertion, while the least negative genetic correlation was −0.010 and occurred between nipple placement and rump width and nipple diameter and mobility. The standard errors associated with the genetic correlations were low and negligible (µ = 0.0001), with the highest error associated with rear legs rear view.

Phenotypic correlations values were low to high and ranged from −0.230 to 0.450, with standard errors being 0.0001 on average. The highest positive phenotypic correlation (0.450) occurred between rump width and chest width while the lowest positive genetic correlation occurred between rump angle and bone quality (0.010). The only variable pair which did not genetically correlate was that comprising nipple diameter and body depth.

The highest negative phenotypic correlation was −0.210 and occurred between rear insertion height and chest width, while the lowest negative phenotypic correlation was −0.010 and occurred between rear legs rear view and udder depth. Mobility and stature, mobility and chest width, between rear legs side view and nipple placement, rear legs side view and udder depth, rear legs side view and rump angle, nipple diameter and rump angle, and rump angle and udder depth were neither positively nor negatively phenotypically correlated.

## 4. Discussion

### 4.1. Heritabilities for Zoometric/LAS Traits and Their Evolution

As reported by Fernández Álvarez et al. [21], the Murciano-Granadina breed has experienced an average gain in heritability values for zoometrics/LAS of 0.1082 and an average decrease in standard errors of 0.0706 (Appendix A) since 2011. Unlike previous genetic evaluations, convergence was attained for all zoometric/LAS traits. Overall, the heritability estimates observed agreed with those in the literature by Manfredi et al. [15], Rupp et al. [31], McLaren et al. [22] and Luo et al. [32]. However, estimates observed most closely resembled those in Manfredi et al. [15], Rupp et al. [33] and Luo et al. [34]. The reason for this may be the fact that McLaren et al. [20] used mixed breed individuals in the genetic evaluations that the authors performed.

Although the average heritability of zoometric measures has reportedly been described as higher than the heritability of LAS traits and usually present larger standard errors [34], the process of validation and optimization of the scales used and the implementation of the system by trained operators makes both parameters equal [4,15].

Indeed, such dissimilarities may occur when LAS scale units are not able to represent the same range of units found in the population for a particular zoometric trait, thus LAS is unable to capture all the population’s variability for such traits. This normally occurs due to the lack of implementation of a process of scale validation and optimization, and trained score operators not being used to collect the information as it has occurred in the Murciano-Granadina goat breed [4].

The progressive gain in heritability values and reduction in heritability standard errors may be ascribed to the technification and improvement of the efficiency of the methods used to collect either phenotypic or genealogic data in terms of quantity and quality. Relative increases in heritability may evidence faster trait evolution; this means fewer generations may be required for traits to evolve either positively or negatively. According to Haworth et al. [35], heritability increases as more genes come into play as individuals undergo major transitions. Our study suggests some of those increases/decreases may reflect underlying changes in the bodies of does as they go through their first parity, and as they accumulate further parities along their lives [36] or in bucks when they periodically go through rutting [37]. Rutting event occurrence varies along the year depending on the breed, although normally it occurs around autumn.

When compared to the values in Fernández Álvarez et al. [21], mobility was the only trait for which a loss in heritability (+0.0500) was reported after a decade. Interestingly, such a loss was parallel to the reduction of −0.1000 in heritability standard errors. Indeed, the first may be a consequence of the second, as the drastic reduction of standard errors may indeed be an increase in the accuracy of this parameter estimation. This may derive from the optimization of the methods used to assess mobility, which on a regular basis are not standardized and may rather depend on the degree of objectivity of the criteria and training of operators. Moreover, Fernández Álvarez et al. [11] reported a reduction in the scale for mobility from 1 to 9 points to 1 to 5 which may have stemmed from the fact that Murciano-Granadina does’ and bucks’ mobility may not describe such a diverse range of mobility values as to cover all the levels in the scale that was formerly applied, which in turn adds to the reduction in heritability for the mobility trait. Furthermore, the value of heritability can change if the impact of environment (or of genes) in the population is substantially altered, for example, as farms implement improvement in their management or systems of phenotypic productive data collection [38]. In this context, if the environmental variation encountered by different individuals increases, then the heritability figure decreases.

Changes in heritability must be regarded with caution given heritability does not measure the proportion of a trait caused by genes, but the proportion of variation in a trait that can be attributed to genes. As a result, when the environment relevant to a given trait uniformly changes affecting all members of the population, that is, the variation or differences among individuals in the population remains the same, the mean value of the trait will change without any change in its heritability. This not only becomes evident for traits for which convergence had not been reached at previous evaluations (body depth, rump angle, bone quality, udder width and rear legs side view), but for those, such as stature, which accounts with a high heritability of 0.4300, even if average stature continues to increase through the years to reach the international optimal standard for the dairy goat type [21]. This means high heritabilities may not necessarily mean that average group differences may ascribe to genes, but to the relationship of those genes with environment, which is of extreme relevance in locally adapted breeds following a process of selection towards a particular productive outcome, such as Murciano-Granadina.

Total phenotypic variance is the denominator of heritability and it is estimated as the variance of the trait being evaluated after correcting for known fixed effects such as sex, and covariates such as age, as it occurred in this study. As extended among animal breeders, the best prediction of future performance is obtained by considering the amount of variation that is not accounted for by known environmental effects. The lack of knowledge in regards these factors increases the estimates of phenotypic variance thus reduces the estimate of heritabilities. However, zoometry needs to follow a rather evolutionary perspective and focus on the total variation between individuals.

Visscher et al. [39], suggested the prediction of the response to selection of specific traits, such as zoometrics/LAS ones, depends on whether selection takes place within or across the factors that cause variation, for instance, year-to year fluctuations within and among herds. Even with the thorough consideration of other factors such as climate and diet, among others, that presumably have a large effect on mean zoometrics, all factors have not been considered. This is due to the fact that selection practices operate at a farm/herd level and within years. Consequently, the best prediction of response would be based on a heritability that is estimated by adjusting for farm between-year variation rather than other factors which may initially be stronger conditioners of zoometrics.

The highest estimates were generally associated with the udder- and teat-related traits, whereas those estimated for the legs and feet were lower. The highest estimates were generally associated with the stature, the udder (nipple-related traits) and those traits involved in teat suspensory system, whereas those estimated for mobility, legs, feet and other body areas which are involved in movement development were the lowest ones. The individual traits with the overall highest and lowest heritability estimates were stature (0.4300) and rear legs side view (0.0906), respectively. Other authors [15,22] have also reported higher estimates for the udder and teat traits compared with the legs and feet in general; even despite using a similar scale and scoring system, some of the traits considered were not the same as those in the present study.

The evaluation of legs aplomb-related traits in Murciano-Granadina goats (rear legs side and rear views and mobility) reported heritability values ranging from 0.0906 to 0.2213. This range covers the values reported for standardized breeds (0.16 and 0.12 for the Alpine and Saanen breeds, respectively) and was also reasonably similar to those of 0.13 reported by McLaren et al. [22] for random crossings between British Alpine, Saanen and Toggenburg. However, heritability values were centred around the middle of the range and were neither as high or low as any of the range limits in the present study.

This situation may be ascribed to zoometric/LAS criteria differing in terms of the methods that were implemented for their collection (different scales or even different trait definition). Furthermore, the rather advanced level of standardization of the breeds that were evaluated may be a source for reduction in the variability for specific zoometric/LAS traits.

When the same or similar scoring methods are used, values closely resemble those in our study (0.2213), as shown by the 0.21 heritability values presented by Luo et al. [32] for a multiracial evaluation involving Alpine, LaMancha, Nubian, Oberhasli, Saanen and Toggenburg goat breeds. The higher heritability reported for rear legs seen from the rear than from the side may derive from the fact that visualization of the area is easier hence, the ability of operator to detect representative animals for a wider range of the scale is feasible. As suggested by McLaren et al. [22] on-farm previous selection criteria may have only selected animals with better aplomb and mobility patterns to remain in the herd, which determines the relative fixation of traits such as rear legs side view and the reduction in the variability and in turn of the heritability of the trait.

Fernández Álvarez et al. [11,21] suggested the fact that the traits comprised within the legs and aplomb major area may have experienced a drastic improvement in terms of the efficiency with which the variability in the scale represents the variability perceived on field, but also of the ability of the operators involved in measure collection to capture such a variability across the levels of each particular scale for each trait [10].

Considering the mammary system major area, the estimates observed in the present study were in close agreement with those observed by Manfredi et al. [15] and Rupp et al. [33] in Saanen and Alpine breeds, Mavrogenis et al. [40] in the Damascus breed, and Luo et al. [32] in a multiracial evaluation involving Alpine, LaMancha, Nubian, Oberhasli, Saanen and Toggenburg goat breeds. Heritabilities for mammary system major area traits ranged from 0.1000 to 0.4100 for udder width and nipple diameter, respectively. Nipple diameter and location had heritabilities of 0.2700 to 0.4100, respectively. These heritability values are in the range of the studies by Mavrogenis et al. [40] and Luo et al. [32] who found intermediate heritabilities over 0.35 for udder and teat characteristics (Table 4).

Although the results in this study are in the range of the aforementioned studies [14,22,32,33,41], noticeable differences are present. The largest of such differences may be ascribed to the number of zoometric/LAS data records available for primipara and multipara does (n  =  39,838). Furthermore, the Murciano-Granadina breed routinely follows a parentage DNA testing of the animals in the kinship matrix. The kinship matrix comprises a total of 279,264 individuals. The previously discussed studies performed genetic evaluations using data from slightly lower than 19,000 to slightly over 43,000 does recorded over several years. Our study considered the information from 39,838 multipara and primipara does, which is in the upper limit of data used in previous research experiences.

McLaren et al. [22], suggested the accuracy of heritability estimates drastically increases as more information is available, which is particularly supported by the negligible values for standard errors in this study (Table 4).

However, differences may not only derive from accuracy issues. For instance, selection of specific traits may involve a reduction in variability and in turn this may translate into the progressive reduction of heritability estimates. The implementation of on-farm selection policies tends to remove effectives displaying undesirable conformations from an early age before animals are able to disseminate their genetics and become established in the herd. The optimization of zoometric and LAS systems and of the ability of operators to collect information may be responsible for the increases in heritability experienced by almost all traits, as reported in Fernández Álvarez et al. [18], as these often translate into an adequation of the scales used to measure or score animals and thus to capture the variability in the population, but also in an increased ability by operators to perceive differences. The Murciano-Granadina breed is an autochthonous population whose additive genetic variance remains relatively stable over time as a result of breed standardization, with the need for decades for significant changes in heritability estimate to occur [18].

### 4.2. Genetic and Phenotypic Correlations among Zoometric/LAS Traits

Similar values to those genetic correlations found in this study were reported by McLaren et al. [19], who also found feet- and legs-related traits to account for the highest standard errors.

There was a general lack of parallelism in the magnitude of genetic correlations and the respective phenotypic correlations for the same pair of variables. This means that although the moderate to high values of genetic relationship among variable pairs permits the determination of a well-defined relationship between trait pairs in either direction (positive or negative), phenotypic correlations are low to mild. This situation challenges selection if we only consider what we can visually see of zoometric traits, and is typical of breeds which are immersed in a process of standardization such as the Murciano-Granadina [3]. This becomes even more complex when genetic and phenotypic correlations present a different sign, with genetic correlations being positive or negative for a specific pair of traits while the corresponding value for phenotypic correlations describes the opposite trend. As reported in Table 5, this situation occurs in the traits of body depth and less sharply in rear legs rear view and mobility traits. The heritability for these traits is in the lowest end of the range for the heritabilities reported in this study.

In line with these findings, Fernández Álvarez et al. [4] reported evidence of a common data structure for the aforementioned traits which defined the configuration of the category of the “mobility and propulsion system” at the principal component analysis that the authors performed. The same authors would also recommend discarding the trait of rump angle from the panel of traits due to its redundant nature in regards its data variability explanatory potential. According Dyce et al. [41], the basis for such dimensionality relies in the continuity of common aponeurosis of the longissimus dorsi muscle given its implication with the development of back motion, and the middle gluteal muscle given its instrumental role in the mechanisms of propulsion.

### 4.3. Major Area Zoometric/LAS Traits Configuration Assessment

The evaluation of genetic and phenotypic parameters across major areas revealed the set of traits comprised in the structure and capacity major area genetically correlate following a positive pattern, which at the very least duplicates the magnitude of phenotypic correlations. There are only two exceptions to note, which are the low negative phenotypic correlation between stature (height at withers) and body depth against the moderate positive genetic correlation for the same pair of traits, with the latter presenting the lowest heritability of the traits in the structure and capacity major area. This demonstrates the more limited range of possibilities for selection to be efficient and the existence of redundancies based on the rump angle trait as evidenced by the studies of Fernández Álvarez et al. [4]. These also presented a low heritability of 0.1706 and a relatively higher standard error of prediction for the rest of traits in the same major area.

The previously described genetic/phenotypic pattern is also shared by the angularity trait of the dairy structure major area. Contrastingly, angularity and bone quality, both from the dairy structure major area describe an opposite relationship. This means that our focus seeks the improvement in one of them, and at the same time we hinder the selection in favour of the other.

As a consequence, the first suggestion is to discard of the rump angle variable from the structure and capacity major area and to include the angularity trait, which may ensure all traits in the same major area behave similarly, which in turn may enhance the potential of selection strategies. This becomes even more important when these results are compared to those in the literature given the same pattern sustained across research experiences in the topic across goat breeds [15,22,33,37]. This may stem from the high values for heritabilities of these traits but also in the objectivity with which such traits can be collected, which configures the solidity of data.

As suggested by our results, the high negative genetic correlations of bone quality and rear insertion height with almost all of the rest of traits makes of these traits potential candidates to be used as references in negative selection practices. In this sense, quantitatively selecting against thicker bones and higher rear insertion heights, which indirectly means selecting for animals with finer and flatter bones, with shorter rear insertion heights (which is the optimum that breeders seek) may indeed result in the optimization of selection practices in the rest of traits, specially of those in the structure and capacity and mammary system major areas.

Within the mammary system major area, rear insertion height described the exact opposite genetic and phenotypic correlation patterns to the rest of traits in the same major area. This stems from the fact that, while performing the extrapolation of zoometrics to LAS, the optimum in LAS scale (9 points in the former and 5 in the new proposal) corresponds to fewer centimetres (3 cm) than the minimum level in the scale (1 point and 11 cm). This may need to be considered when implementing selection strategies as although distribution properties and descriptive statistics are equivalent, the direction of the correlations is inversed when compared to the rest of traits in the same major area. The proposal would be to invert the current LAS scale so that upper levels in the LAS scale correspond to longer rear insertion heights, hence making the LAS system and zoometrics follow the same direction, without the need to remove the trait from the major area in which it has traditionally been evaluated.

This has also been described in other breeds, as CAPRIGRAN LAS is based upon the system which was traditionally applied in international goat breeds such as Alpine and Saanen breeds, as a direct extrapolation from dairy cows, in respect to the direction of the relationships among trait pairs that were considered a priori. In these regards, Manfredi et al. [16] indicated that in goats, as the strength of the medial ligament changed, there is a negative knock-on effect on the angle and placement of the nipples. Thus, selection against “baggy” udders (which means udder traits scoring low) would translate into an indirect response towards bigger, close-in and inner oriented teats. This event may also be the source for the high negative correlation between udder depth and anterior insertion. As we go lower on the LAS scale for anterior insertion, we approach wider angles that reach a minimum of 45° and correspond with 1 on the LAS scale, while the optimum was formerly placed at 9, or currently placed at 5 in the new proposal for the LAS scale and corresponds with 120°. Wider angles imply shallower udders derived from these implanting more cranially in the body of does. For this, the inversion of the scale would permit correlations to agree with the patterns described by the rest of parameters without impairing the aim of selection for anterior insertion.

In this sense, if sturdy, tall, thick-boned animals presenting sloping rumps are selected against, we may indirectly seek for rather average-sized fine-boned animals with raised rumps, which is the exact opposite to that recommended in the literature for Holstein Friesian dairy cows [42,43], but which in goats maximizes the space between hocks which in turn means goats present broader spaces for the mammary system to be installed.

In regards the legs aplomb major area, all traits considered (rear legs side and rear views and mobility) describe the similar negative genetic correlation trend with the rest of the traits in the rest of the major areas, which permits the consideration of these traits as a solid cluster in terms of the planification of selection strategies, even if the direction of selection must be the opposite to that to be performed for the structure and capacity and mammary system major areas.

## 5. Conclusions

Changes occurring along lactations in Murciano-Granadina does and during rutting in Murciano-Granadina bucks need to be accounted for while measuring individuals due to the zoometric alterations that they promote. Certain traits may not be able to account for the variability described in international 9-point scoring scales, but a 5-point scale with the consequent reduction of heritability may be an improvement. A priori on-farm selection criteria may have selected animals with better aplomb and mobility patterns, which is the source for their relative fixation in the population, variability and heritability reduction. Discarding the rump angle trait from the structure and capacity major area and including the angularity trait ensures all traits in the same major area behave similarly, which enhances the potential of selection strategies. Scale inversion on specific zoometric traits may help to address disagreement in the patterns of the rest of the traits in the same major area without impairing the aim of selection for the trait whose scale has been modified. The legs aplomb major area conforms to a solid cluster in terms of the planification of selection strategies, even if the direction of selection must be the opposite to that for the structure and capacity and mammary system major areas. Future breeding programs would benefit from modifying the collection system and the manner in which the zoometric traits are managed at a genetic level to ensure that selection for zoometrics/LAS does not translate into any unwanted change in functional fitness, maximizing the outcome of selection strategies to fit the particular reality of the goat species and the diverse range of breeds that it comprises.

## Figures and Tables

**Table 1 animals-13-01114-t001:** Detailed description of the scales used and the translation process from zoometric traits to LAS scores in Murciano-Granadina goat and bucks.

Gender/ Status	Major Area	Linear Trait	Zoometric Scale/Categorical Scale	Zoometric Optimum Scoring	Reference/Middle Point	LAS Extrapolation	LAS Optimum Scoring	New Scale Proposal [14]
Primipara/ Multipara does	Structure and capacity	Stature (Height to withers)	62–78 cm	72 cm (primipara) and 74 cm (multipara)	5 (70 cm)	1–9 points	6 (primipara) and 7 (multipara)	1–9 points
		Chest Width	15–23 cm	20 cm (primipara) and 21 cm (multipara)	5 (19 cm)	1–9 points	6 (primipara) and 7 (multipara)	1–9 points
		Body Depth	Shallow–Extremely deep	Intermediate	5 (elbow end matches rib depth)	1–9 points	7 (primipara and multipara)	1–8 points
		Rump Width	13–21 cm	18 cm (primipara) and 19 (multipara)	5 (17 cm)	1–9 points	6 (primipara) and 7 (multipara)	1–7 points
		Rump Angle	55°–31°	31°	5 (43°)	1–9 points	9	1–7 points (Not relevant) [4]
	Dairy structure	Angularity	Angulous extremity–Rough extremity	Angulous extremity	5 (Intermediate)	1–9 points	9	1–10 points
		Bone Quality	Round and rough bones–flat and neat bones	Flat and neat bones	5 (Intermediate)	1–9 points	9	1–5 points
	Mammary system	Anterior insertion	Weak–Strong	120º	5 (90°)	1–9 points	9	1–5 points
		Rear Insertion Height	11–3 cm	3 cm	5 (7 cm)	1–9 points	9	1–5 points
		Median Suspensor Ligament	1–9 cm	5 cm	5 (5 cm)	1–9 points	5	1–6 points
		Udder width	3–11 cm	11 cm	5 (7 cm)	1–9 points	9	1–5 points
		Udder Depth	−10–10 cm	−5 cm (5 cm over hock level) and 0 cm (udder bottom at hock level)	5 (0 cm/at hock level)	1–9 points	3 (primipara) and 5 (multipara)	1–9 points
		Nipple placement	90°–0°	0°	5 (45°)	1–9 points	9	1–6 points
		Nipple Diameter	0.5° to 4.5°	2 cm	5 (2.5 cm)	1–9 points	4	1–9 points
	Legs aplomb	Rear Legs Rear View	Very close–Parallel and separated	Parallel and separated	5 (slightly close)	1–9 points	9	1–7 points
		Rear Legs Side View	Straight–Very curved	Desirable curvature. A short distance from an imaginary line to anterior curvature of hock	5 (desirable curvature)	1–9 points	5	1–7 points
		Mobility	Very bad mobility due to skeleton structure–long and strong, straight and uniform stride	Good mobility. Easy and harmonic movement	5 (moderate mobility)	1–9 points	9	1–5 points
Bucks	Structure and capacity	Stature (Height to withers)	68–92 cm	83 cm (young) and 86 cm (adult)	5 (80 cm)	1–9 points	6 (bucklings) and 7 (bucks)	1–10 points
		Chest Width	15–31 cm	25 cm (young) and 27 cm (adult)	5 (23 cm)	1–9 points	6 (bucklings) and 7 (bucks)	1–11 points
		Body Depth ^a^	Shallow–Extremely deep	Intermediate	5 (elbow end matches rib depth)	1–9 points	7 (bucklings and bucks)	1–7 points
		Rump Width	14–22 cm	19 cm (young) and 20 cm (adult)	5 (18 cm)	1–9 points	6 (bucklings) and 7 (bucks)	1–5 points
		Rump Angle	55–31°	31°	5 (43°)	1–9 points	9	1–6 points
	Dairy structure	Angularity ^a^	Angulous extremity–Rough extremity	Angulous extremity	5 (Intermediate)	1–9 points	9	1–9 points
		Bone Quality ^a^	Round and rough bones–flat and neat bones	Flat and neat bones	5 (Intermediate)	1–9 points	9	1–5 points
	Legs aplomb	Rear Legs Rear View ^a^	Very close–Parallel and separated	Parallel and separated	5 (slightly close)	1–9 points	9	1–6 points
		Rear Legs Side View ^a^	Straight–Very curved	Desirable curvature. Short distance from an imaginary line to anterior curvature of hock	5 (desirable curvature)	1–9 points	5	1–7 points
		Mobility ^a^	Very bad mobility due to skeleton structure–long and strong, straight and uniform stride	Good mobility. Easy and harmonic movement	5 (moderate mobility)	1–9 points	9	1–5 points

^a^ Same criteria for bucks and does.

**Table 2 animals-13-01114-t002:** Type III Sum of Squares analysis for zoometric traits to LAS scores in Murciano-Granadina primipara and multipara does (41,323) and bucks (1485).

Major Area	Linear Trait	Degrees of Freedom (df)	F	Pr > F	Error DF	Residual Sum of Squares (RSS)	Residual Mean Squares (RMS)
Structure and capacity	Stature (Height to withers)	490	186.36	<0.0001	40,832	223,454.22	5.47
	Chest Width	490	194.49	<0.0001	40,832	53,037.59	1.30
	Body Depth	490	86.68	<0.0001	40,832	28,484.34	0.70
	Rump Width	490	73.71	<0.0001	40,832	18,454.07	0.45
	Rump Angle	490	22.39	<0.0001	40,832	244,385.19	5.99
Dairy structure	Angulosity	490	54.49	<0.0001	40,832	3,602,289.62	88.22
	Bone Quality	490	64.25	<0.0001	40,832	19,497.63	0.48
Mammary system	Anterior insertion	418	20.36	<0.0001	39,419	21,571.54	0.55
	Rear Insertion Height	418	45.46	<0.0001	39,419	27,288.45	0.69
	Median Suspensor Ligament	418	35.96	<0.0001	39,419	52,341.04	1.33
	Udder width	418	59.41	<0.0001	39,419	27,976.58	0.71
	Udder Depth	418	68.59	<0.0001	39,419	342,127.71	8.68
	Nipple placement	418	12.86	<0.0001	39,419	2,869,493.07	72.79
	Nipple Diameter	418	10.71	<0.0001	39,419	20,827.55	0.53
Legs aplomb	Rear Legs Rear View	490	77.71	<0.0001	40,832	16,404.11	0.40
	Rear Legs Side View	490	83.17	<0.0001	40,832	17,808.44	0.44
	Mobility	490	20.94	<0.0001	40,832	14,415.85	0.35

**Table 3 animals-13-01114-t003:** Goodness of fit coefficients of the model testing for zoometric traits to LAS scores in Murciano-Granadina primipara and multipara does (41,323) and bucks (1485).

Major Area	Linear Traits	Degrees of Freedom (df)	Predicted R^2^	Adjusted R^2^	Mean Squared Error (MSE)	Root Mean Squared Error (RMSE)	Durbin Watson (DW)
Structure and capacity	Stature (Height to withers)	40,832	0.691	0.687	5.473	2.339	1.691
	Chest Width	40,832	0.700	0.696	1.299	1.140	1.575
	Body Depth	40,832	0.510	0.504	0.698	0.835	1.446
	Rump Width	40,832	0.469	0.463	0.452	0.672	1.857
	Rump Angle	40,832	0.212	0.202	5.985	2.446	1.787
Dairy structure	Angulosity	40,832	0.395	0.388	88.222	9.393	1.742
	Bone Quality	40,832	0.435	0.429	0.478	0.691	1.747
Mammary system	Anterior insertion	39,419	0.178	0.169	0.547	0.740	1.911
	Rear Insertion Height	39,419	0.325	0.318	0.692	0.832	1.805
	Median Suspensor Ligament	39,419	0.276	0.268	1.328	1.152	1.901
	Udder width	39,419	0.387	0.380	0.710	0.842	1.564
	Udder Depth	39,419	0.421	0.415	8.679	2.946	1.890
	Nipple placement	39,419	0.120	0.111	72.795	8.532	1.926
	Nipple Diameter	39,419	0.102	0.092	0.528	0.727	1.924
Legs aplomb	Rear Legs Rear View	40,832	0.483	0.476	0.402	0.634	1.888
	Rear Legs Side View	40,832	0.500	0.493	0.436	0.660	1.747
	Mobility	40,832	0.201	0.191	0.353	0.594	1.902

**Table 4 animals-13-01114-t004:** Estimated components of variance, heritability (h^2^) and standard error of the mean (SEM) for zoometric and LAS traits obtained from multivariate analyses through REML methods.

Major Area	Zoometric/LAS Trait	$\sigma_{a}^{2}$ ± SEM	$\sigma_{p}^{2}$ ± SEM	$\sigma_{e}^{2}$ ± SEM	h^2^ ± SEM
Structure and capacity	Stature (Height to withers)	0.4986 ± 0.0002	1.15511 ± 0.00006	0.6565 ± 0.0002	0.4300 ± 0.0001
	Chest Width	0.3094 ± 0.0003	1.05539 ± 0.00006	0.7460 ± 0.0002	0.2906 ± 0.0025
	Body Depth	0.0666 ± 0.0002	0.67957 ± 0.00002	0.6130 ± 0.0002	0.1000 ± 0.0001
	Rump Width	0.1370 ± 0.0001	0.43747 ± 0.00003	0.3005 ± 0.0001	0.3100 ± 0.0001
	Rump Angle	0.1096 ± 0.0001	0.63168 ± 0.00003	0.5221 ± 0.0001	0.1706 ± 0.0025
Dairy structure	Angulosity	0.2699 ± 0.0001	1.06034 ± 0.00002	0.7904 ± 0.0001	0.2513 ± 0.0034
	Bone Quality	0.1479 ± 0.0001	0.47679 ± 0.00002	0.3289 ± 0.0001	0.3100 ± 0.0001
Mammary system	Anterior insertion	0.1176 ± 0.0002	0.54863 ± 0.00007	0.4310 ± 0.0002	0.2106 ± 0.0025
	Rear Insertion Height	0.1691 ± 0.0003	0.65676 ± 0.00005	0.4877 ± 0.0002	0.2588 ± 0.0034
	Median Suspensor Ligament	0.3758 ± 0.0003	1.13703 ± 0.00006	0.7612 ± 0.0002	0.3300 ± 0.0001
	Udder width	0.0515 ± 0.0001	0.52287 ± 0.00002	0.4714 ± 0.0001	0.1000 ± 0.0001
	Udder Depth	0.4014 ± 0.0002	1.37782 ± 0.00004	0.9764 ± 0.0002	0.2900 ± 0.0001
	Nipple placement	0.1533 ± 0.0002	0.56946 ± 0.00003	0.4162 ± 0.0001	0.2700 ± 0.0001
	Nipple Diameter	0.8757 ± 0.0002	2.13335 ± 0.00006	12576 ± 0.0002	0.4100 ± 0.0001
Legs aplomb	Rear Legs Rear View	0.0883 ± 0.0005	0.40271 ± 0.00012	0.3144 ± 0.0004	0.2213 ± 0.0050
	Rear Legs Side View	0.0379 ± 0.0004	0.42775 ± 0.00012	0.3899 ± 0.0004	0.0906 ± 0.0025
	Mobility	0.0393 ± 0.0001	0.35306 ± 0.00001	0.3138 ± 0.0001	0.1100 ± 0.0001

**Table 5 animals-13-01114-t005:** Estimated genetic (r_G_) (above diagonal) and phenotypic (r_P_) (below diagonal) correlations for zoometric and LAS traits obtained in bivariate analyses through REML.

	Major Area	Structure and Capacity					Dairy Structure		Mammary System							Legs Aplomb		
Major Area	Linear Traits	Stature (Height to Withers)	Chest Width	Body Depth	Rump Width	Rump Angle	Angularity	Bone Quality	Anterior Insertion	Rear Insertion Height	Median Suspensor Ligament	Udder width	Udder Depth	Nipple Placement	Nipple Diameter	Rear Legs Rear View	Rear Legs Side View	Mobility
Structure and capacity	Stature (Height to withers)		0.530	0.220	0.610	0.050	0.320	−0.460	−0.230	−0.400	0.150	0.120	0.080	−0.160	0.070	−0.340	−0.130	−0.360
	Chest Width	0.340		0.620	0.790	0.280	0.700	−0.490	0.070	−0.500	0.140	0.230	0.070	0.040	0.090	−0.120	−0.110	−0.150
	Body Depth	−0.030	0.260		0.530	0.150	0.420	−0.420	0.100	−0.370	0.050	0.090	0.110	−0.040	0.000	−0.210	0.160	−0.130
	Rump Width	0.290	0.450	0.210		0.260	0.560	−0.530	0.070	−0.440	0.130	0.260	0.000	−0.010	0.080	−0.140	−0.030	−0.200
	Rump Angle	0.050	0.140	0.050	0.140		0.300	0.010	0.230	−0.320	0.030	0.160	−0.060	0.090	0.040	0.250	0.080	0.150
Dairy structure	Angulosity	0.160	0.430	0.250	0.370	0.150		−0.320	0.210	−0.390	0.100	0.320	0.090	0.130	0.060	−0.030	−0.290	−0.110
	Bone Quality	−0.120	−0.180	−0.070	−0.150	−0.040	−0.120		0.140	0.420	−0.110	−0.040	−0.120	0.110	−0.060	0.380	0.050	0.440
Mammary system	Anterior insertion	0.060	0.120	0.070	0.120	0.120	0.150	−0.030		0.240	−0.110	0.160	−0.570	0.210	−0.120	0.300	0.020	0.350
	Rear Insertion Height	−0.110	−0.210	−0.040	−0.160	−0.150	−0.160	0.070	−0.070		−0.050	0.150	−0.190	0.150	−0.020	0.280	0.030	0.350
	Median Suspensor Ligament	0.040	0.060	0.030	0.040	0.010	0.060	0.020	−0.090	−0.060		0.130	0.360	0.370	0.320	−0.050	−0.100	−0.100
	Udder width	0.090	0.160	0.170	0.190	−0.040	0.150	−0.020	0.010	0.090	0.050		−0.090	0.320	0.060	0.280	−0.130	0.260
	Udder Depth	0.030	0.100	0.050	0.060	0.000	0.080	−0.010	−0.230	−0.030	0.320	0.130		−0.170	0.240	−0.220	−0.100	−0.370
	Nipple placement	0.010	0.020	0.040	0.050	0.020	0.050	−0.020	0.080	0.030	0.180	0.100	0.050		0.380	0.290	0.090	0.290
	Nipple Diameter	0.080	0.080	0.030	0.080	0.000	0.040	−0.050	−0.060	0.010	0.120	0.100	0.160	0.140		0.020	−0.040	−0.010
Legs aplomb	Rear Legs Rear View	0.040	0.020	0.010	0.030	0.020	−0.030	0.020	0.020	0.040	0.020	0.130	−0.010	0.080	0.050		0.370	0.870
	Rear Legs Side View	−0.070	−0.090	0.010	−0.060	0.000	−0.070	0.050	0.010	0.050	0.010	−0.030	0.000	0.000	0.010	0.030		0.200
	Mobility	0.000	0.000	0.040	0.040	0.050	0.010	0.010	0.070	−0.010	−0.020	0.050	−0.080	0.050	−0.020	0.200	0.130	

Average standard error for estimated genetic (rG) (above diagonal) and phenotypic (rP) were both 0.0001.

## Data Availability

Data will be made available from corresponding author upon reasonable request.

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
