# Peer review of "Analysis of the Genetic Parameters for Dairy Linear Appraisal and Zoometric Traits: A Tool to Enhance the Applicability of Murciano-Granadina Goats Major Areas Evaluation System"

_animals, 2023, doi:10.3390/ani13061114_

Round 1

Reviewer 1 Report (New Reviewer)

GENERAL COMMENT:

I consider this work is within the scope of “Animals”. It contains information useful in a field in which available information is of interest to improve knowledge on dairy goat genetics and selection. Overall, it is well written and organised. I indicate below only minor points to be improved in the manuscript.

TITLE:

It is all right.

ABSTRACT:

It is OK.

KEYWORDS:

These are OK.

INTRODUCTION:

Overall, this section is OK.

Line 53: Type “in vivo” in italics.

Line 62: Insert space at “2010[3].”, thus resulting in “2010 [3].

Near Line 62: The first time you speak about Murciano-Granadina breed, it is advisable to add a bibliographic citation of the breed description, in order to put in context potential reader not familiar with this breed.

Line 69: Add “Spain” at “Andalusia”, thus resulting in “Andalusia (Spain)”. In this manner, international readers can understand better its location.

Check the numbering of bibliographic citations in the text, because these skip from number 20 (on line 20 on page 8) to number 23 (on line 59 on page 9). Therefore, bibliographic quotes 20 and 21 have been skipped.

Lines 112-113: Add a bibliographic citation supporting that “heritability of 0.15 or higher is accepted as indicating at least moderate heritability of a trait”.

Lines 129-133: Add a bibliographic citation supporting this information.

Lines 144-150: Indicate that this objective is for Murciano-Granadina breed (for example as follows: “potential of Murciano-Granadina breed selection strategies is maximized”).

MATERIALS AND METHODS:

Lines 160-165: Provide references or web references for “Order of 22 June 2018” and “Order AYG/287/2019 of 28 of February of 2019” in order for making these orders easy to find by international readers not familiar with local laws.

Lines 35-40 in page 8: Add a bibliographic citation supporting this information.

RESULTS SECTION:

Overall, this section is OK.

Line 160, page 11: remove comma from “Fernández Álvarez, et al. [14]”, thus resulting in “Fernández Álvarez et al. [14]

DISCUSSION SECTION:

This section is OK.

Line 23, page 15: remove comma from “Fernández Álvarez, et al. [19]”, thus resulting in “Fernández Álvarez et al. [19]

The same correction to do in the references with “et al.” that follows from Line 27, page 15.

CONCLUSIONS:

Indicate that these conclusions refer to Murciano-Granadina breed.

AUTHORS CONTRIBUTIONS:

Lines 300-313: according to journal usage, it is better to describe the contribution roles as follows:

“Conceptualization, X.X. and Y.Y.; methodology, X.X.; software, X.X.; validation, X.X., Y.Y. and Z.Z.; formal analysis, X.X.; investigation, X.X.; resources, X.X.; data curation, X.X.; writing—original draft preparation, X.X.; writing—review and editing, X.X.; visualization, X.X.; supervision, X.X.; project administration, X.X.; funding acquisition, Y.Y. All authors have read and agreed to the published version of the manuscript.” You can see the CRediT taxonomy for the term explanation.

REFERENCES SECTION:

In general terms, this section is well organised and adjusted to the style of the journal for references. However, I recommend reviewing it for typos. For example:

Line 359: “Spetember

Line 393: Latin names of the organisms must be written in italics: “Equus asinus”, “Homo sapiens”…

You write, indifferently, the first letter of each word of the article titles as uppercase and as lowercase letter. Please see Instructions for authors and recent articles of the journal to see the style used, in order for homogenising it.

TABLES:

Tables are OK.

Author Response

Reviewer 1

Comments and Suggestions for Authors

GENERAL COMMENT:

I consider this work is within the scope of “Animals”. It contains information useful in a field in which available information is of interest to improve knowledge on dairy goat genetics and selection. Overall, it is well written and organised. I indicate below only minor points to be improved in the manuscript.

Response: We thank the reviewer for his/her kind comments.

TITLE:

It is all right.

Response: We thank the reviewer for his/her kind comments.

ABSTRACT:

It is OK.

Response: We thank the reviewer for his/her kind comments.

KEYWORDS:

These are OK.

Response: We thank the reviewer for his/her kind comments.

INTRODUCTION:

Overall, this section is OK.

Response: We thank the reviewer for his/her kind comments.

Line 53: Type “in vivo” in italics.

Response: We changed it.

Line 62: Insert space at “2010[3].”, thus resulting in “2010 [3].”

Response: We changed it.

Near Line 62: The first time you speak about Murciano-Granadina breed, it is advisable to add a bibliographic citation of the breed description, in order to put in context potential reader not familiar with this breed.

Response: We added it.

Line 69: Add “Spain” at “Andalusia”, thus resulting in “Andalusia (Spain)”. In this manner, international readers can understand better its location.

Response: We added it.

Check the numbering of bibliographic citations in the text, because these skip from number 20 (on line 20 on page 8) to number 23 (on line 59 on page 9). Therefore, bibliographic quotes 20 and 21 have been skipped.

Response: We added it.

Lines 112-113: Add a bibliographic citation supporting that “heritability of 0.15 or higher is accepted as indicating at least moderate heritability of a trait”.

Response: We added it.

Lines 129-133: Add a bibliographic citation supporting this information.

Response: We added it.

Lines 144-150: Indicate that this objective is for Murciano-Granadina breed (for example as follows: “potential of Murciano-Granadina breed selection strategies is maximized”).

Response: We added it.

MATERIALS AND METHODS:

Lines 160-165: Provide references or web references for “Order of 22 June 2018” and “Order AYG/287/2019 of 28 of February of 2019” in order for making these orders easy to find by international readers not familiar with local laws.

Response: We added them.

Lines 35-40 in page 8: Add a bibliographic citation supporting this information.

Response: We added it.

RESULTS SECTION:

Overall, this section is OK.

Response: We thank the reviewer for his/her kind comments.

Line 160, page 11: remove comma from “Fernández Álvarez, et al. [14]”, thus resulting in “Fernández Álvarez et al. [14]”

Response: Comma was removed.

DISCUSSION SECTION:

This section is OK.

Response: We thank the reviewer for his/her kind comments.

Line 23, page 15: remove comma from “Fernández Álvarez, et al. [19]”, thus resulting in “Fernández Álvarez et al. [19]”

Response: Comma was removed.

The same correction to do in the references with “et al.” that follows from Line 27, page 15.

Response: Comma was removed.

CONCLUSIONS:

Indicate that these conclusions refer to Murciano-Granadina breed.

Response: We clarified it.

AUTHORS CONTRIBUTIONS:

Lines 300-313: according to journal usage, it is better to describe the contribution roles as follows:

“Conceptualization, X.X. and Y.Y.; methodology, X.X.; software, X.X.; validation, X.X., Y.Y. and Z.Z.; formal analysis, X.X.; investigation, X.X.; resources, X.X.; data curation, X.X.; writing—original draft preparation, X.X.; writing—review and editing, X.X.; visualization, X.X.; supervision, X.X.; project administration, X.X.; funding acquisition, Y.Y. All authors have read and agreed to the published version of the manuscript.” You can see the CRediT taxonomy for the term explanation.

Response: We corrected it.

REFERENCES SECTION:

In general terms, this section is well organised and adjusted to the style of the journal for references. However, I recommend reviewing it for typos. For example:

Line 359: “Spetember”

Response: We corrected it.

Line 393: Latin names of the organisms must be written in italics: “Equus asinus”, “Homo sapiens”…

Response: We corrected it.

You write, indifferently, the first letter of each word of the article titles as uppercase and as lowercase letter. Please see Instructions for authors and recent articles of the journal to see the style used, in order for homogenising it.

Response: We homogenized it.

TABLES:

Tables are OK.

Response: We thank the reviewer for his/her kind comments.

Reviewer 2 Report (Previous Reviewer 2)

One remark

Page 10, line 178-180: I maintain the suggestion to remove the parentheses from the sentence (Table 2). In the indicated sentence, at the beginning of the sentence, the authors refer to table number 2.

Author Response

Reviewer 2

Comments and Suggestions for Authors

One remark

Page 10, line 178-180: I maintain the suggestion to remove the parentheses from the sentence (Table 2). In the indicated sentence, at the beginning of the sentence, the authors refer to table number 2.

Response: Sorry we had not understood. Now we have deleted it. Thank you for your suggestion.

Reviewer 3 Report (New Reviewer)

Statistical analysis were well written and described. Perfect.

This manuscript can be accepted after major revision.

1. Lines 51-55: Please clearly define what zoometry means for non-familiar readers.

2. Line 63: “type traits”, please give some examples “such as…”.

3. Lines 131-132: “(bucklings younger than two years and bucks of two years old and older) as the most relevant modifications to implement.” For what?

4. Line 157: Animal records were collected in 76 farms” Please, give the Median: Min-Max number of records for farms.

5. In Discussion section: Please mention the probable causes of differences with literature.

6. In Table 3: predicted adjusted coefficient of determination values aren’t so high, please consider it for interpretation.

Author Response

Reviewer 3

Comments and Suggestions for Authors

Statistical analysis were well written and described. Perfect.

Response: We thank the reviewer for his/her kind comments.

This manuscript can be accepted after major revision.

  1. Lines 51-55: Please clearly define what zoometry means for non-familiar readers.

Response: Defined.

  1. Line 63: “type traits”, please give some examples “such as…”.

Response: Added.

  1. Lines 131-132: “(bucklings younger than two years and bucks of two years old and older) as the most relevant modifications to implement.” For what?

Response: Explained.

  1. Line 157: Animal records were collected in 76 farms” Please, give the Median: Min-Max number of records for farms.

Response: Added

  1. In Discussion section: Please mention the probable causes of differences with literature.

 Response: We followed the reviewer suggestion.

  1. In +: predicted adjusted coefficient of determination values aren’t so high, please consider it for interpretation.

 Response: We followed the reviewer suggestion.

Round 2

Reviewer 3 Report (New Reviewer)

Thank you for your valuable corrections.

This manuscript is a resubmission of an earlier submission. The following is a list of the peer review reports and author responses from that submission.

Round 1

Reviewer 1 Report

The topic is interesting, but the total manuscript is extremly long. It definitely has to be rewritten and shortened, especially the Introduction and M&M sections. There are too detailled information in the M&M section, what is not needed in a scientific paper. The detailled descriptions have to be removed, only a short presentation of methods is favourable. (Be careful with numbering of sub-sections, as there are two 2.4 paragraphs...)

Table2 also has to be reconstructed, as SS and MS columns are not needed.

Table3 contains MSE and RMSE, I think RMSE is completely enough.

Conclusion is also too long, some parts should be moved to the Discussion section.

Author Response

Reviewer 1

The topic is interesting, but the total manuscript is extremly long. It definitely has to be rewritten and shortened, especially the Introduction and M&M sections. There are too detailled information in the M&M section, what is not needed in a scientific paper. The detailled descriptions have to be removed, only a short presentation of methods is favourable. (Be careful with numbering of sub-sections, as there are two 2.4 paragraphs...)

Response: We thank the reviewer for his/her kind comments. We followed reviewer suggestions as follows..

Table2 also has to be reconstructed, as SS and MS columns are not needed.

Response: Columns suggested were removed.

Table3 contains MSE and RMSE, I think RMSE is completely enough.

Response: The information that these parameters provide is different. RMSE is the standard deviation of the errors which occur when a prediction is made on a dataset and measures the error of prediction. Hence may be analogous to standard error. That is the difference in difference between observed and predicted values. Thus, relevant in the context of the study that is described in the present paper. Parallelly, MSE may be more affine to the concept of standard deviation that is the difference of values from mean and measures the spread of observations around the mean. The principle on which PBVs relies is that for a fair comparison among animals, we need to consider their observed phenotype as a deviation of an expected mean, hence the relevance of this parameter in the context of the study.

Conclusion is also too long, some parts should be moved to the Discussion section.

Response: We followed the reviewer’s suggestion.

Reviewer 2 Report

The scientific article submitted by the authors provides a lot of information important from the cognitive and application point of view. I am concerned about the way the results are presented, as well as the division of content between chapters. The reviewed article is very extensive in text. It's not easy to read! In my opinion, authors should edit the content of the article, starting from the Introduction through the Conclusions. Authors should limit the size of the article. I believe it is possible without compromising its quality. Much more synthetically: Introduction, Discussion, Conclusions! More synthetically: aim of the research.

In the chapter "Results" there is no descriptipon of the estimated heritability and genetic correlations. I find such a detailed description of some statistical methods and measures too detailed, e.g. VIF, Fisher's F test etc. Attach an additional table with descriptive statistics to the article - the authors write about them on page 11, lines 173-181.

Detailed comments

Page 1; Line 18: LAS - Provide an explanation of the abbreviation.

Page 2; Lines 56-60: Literature source should be included.

Page 4; line 176: “41 481 across the year” or for all years of the research?

Page 5; line 204: Was there really only one person who rated the animals? Was there an assessment of the repeatability of this assessment?

Page 5; Line 222: “Figures S1 to S27” where can I find the figures?

Page 8; Line 3: should be Kolmogorov.

Page 8; Lines 21-32: Personally, I find this passage redundant.

Page 9; Line 81: table 4, check it please.

Page 10: Line 110-111: Repetition from page 4, lines 173-174.

Page 10; 130-145: Correct the description of the statistical model. I suggest listing the individual effects included in the model in the following lines. Check the indices, write its in italic. Yijklmno?

Page 10; line 146: Restricted, write from small letter.

Page 11; line 153: “same software” – unclear.

Page 11; 162-165: “For this… sample size” – remove these sentences.

Page 11; line 194: remove “(Table)”.

Page 12: Table 2: I suggest to remove 2 columns: “Degrees of freedom” and “Sum of squares”.

Page 12; line 206: values: 0.05, 0 – where can I find its.

Page 13; Table 4: from the title, remove a sentence: “Results for zoometric and LAS traits were exactly the same” – move it to the results.

Page 14; table 5: I don’t understand the legend “ASE … were 0.0001”?

Page 15; line 7: “Table S5” – where can I find the information?

Page 18; lines 158-171: Move a big party of the paragraph to the “Results”.

Page 19; lines 226-232: Literature source should be included.

Author Response

Reviewer 2

The scientific article submitted by the authors provides a lot of information important from the cognitive and application point of view. I am concerned about the way the results are presented, as well as the division of content between chapters. The reviewed article is very extensive in text. It's not easy to read! In my opinion, authors should edit the content of the article, starting from the Introduction through the Conclusions. Authors should limit the size of the article. I believe it is possible without compromising its quality. Much more synthetically: Introduction, Discussion, Conclusions! More synthetically: aim of the research.

Response: We thank the reviewer for his/her kind comments. We followed the reviewer’s suggestion.

In the chapter "Results" there is no descriptipon of the estimated heritability and genetic correlations. I find such a detailed description of some statistical methods and measures too detailed, e.g. VIF, Fisher's F test etc. Attach an additional table with descriptive statistics to the article - the authors write about them on page 11, lines 173-181.

Response: We followed the reviewer’s suggestions. Descriptive statistics are present in Supplementary Tables.

Detailed comments

Page 1; Line 18: LAS - Provide an explanation of the abbreviation.

Response: We followed the reviewer’s suggestion.

Page 2; Lines 56-60: Literature source should be included.

Response: This paragraph was removed as suggested by the reviewer.

Page 4; line 176: “41 481 across the year” or for all years of the research?

Response: We corrected it.

Page 5; line 204: Was there really only one person who rated the animals? Was there an assessment of the repeatability of this assessment?

Response: Yes. Only one person. Yes, the person in charge of measuring is the official appraiser for the breed. As stated in the text, animals were only scored once in their lifetime. This person was approved and trained, and this training involved repeatability tests needing for his consolidation as their official, appraiser.

Page 5; Line 222: “Figures S1 to S27” where can I find the figures?

Response: Sorry, we forgot to include them. Now they have been included.

Page 8; Line 3: should be Kolmogorov.

Response: We disagree, it is correct.

Page 8; Lines 21-32: Personally, I find this passage redundant.

Response: We removed it following the reviewer’s suggestion..

Page 9; Line 81: table 4, check it please.

Response: It was revised and corrected.

Page 10: Line 110-111: Repetition from page 4, lines 173-174.

Response: Although numbers are the same, this is not a repetition. The second time it is mentioned it refers to the kinship matrix to perform genetic analyses. Some times these two do not coincide, hence the need to state it.

Page 10; 130-145: Correct the description of the statistical model. I suggest listing the individual effects included in the model in the following lines. Check the indices, write its in italic. Yijklmno?

Response: We followed the reviewer suggestion. We decided not use italics given the particular policy of this journal which is somehow strict about their use.

Page 10; line 146: Restricted, write from small letter.

Response: We followed the reviewer’s suggestion.

Page 11; line 153: “same software” – unclear.

Response: We clarified it.

Page 11; 162-165: “For this… sample size” – remove these sentences.

Response: We removed it as suggested by the reviewer.

Page 11; line 194: remove “(Table)”.

Response: We removed it as suggested by the reviewer.

Page 12: Table 2: I suggest to remove 2 columns: “Degrees of freedom” and “Sum of squares”.

Response: We removed them as suggested by the reviewer.

Page 12; line 206: values: 0.05, 0 – where can I find its.

Response: As stated below in the body text, this information can be consulted in Table S3. However, we clarified this in the body text.

Page 13; Table 4: from the title, remove a sentence: “Results for zoometric and LAS traits were exactly the same” – move it to the results.

Response: We followed the reviewer’s suggestion.

Page 14; table 5: I don’t understand the legend “ASE … were 0.0001”?

Response: We corrected it. This means estimation error for each correlation was 0.0001.

Page 15; line 7: “Table S5” – where can I find the information?

Response: In Supplementary Material.

Page 18; lines 158-171: Move a big party of the paragraph to the “Results”.

Response: We followed the reviewer’s suggestion.

Page 19; lines 226-232: Literature source should be included.

Response: This information comes from our results. From the interpretation of our results. We clarified this in the body text.